# Identification of EZH2 as Cancer Stem Cell Marker in Clear Cell Renal Cell Carcinoma and the Anti-Tumor Effect of Epigallocatechin-3-Gallate (EGCG)

**DOI:** 10.3390/cancers14174200

**Published:** 2022-08-30

**Authors:** Chen Lyu, Lili Wang, Birgit Stadlbauer, Elfriede Noessner, Alexander Buchner, Heike Pohla

**Affiliations:** 1Tumor Immunology Laboratory, LIFE Center, LMU Klinikum, University Munich, D-82152 Planegg, Germany; 2Department of Radiology, First Affiliated Hospital, Zhejiang University School of Medicine, Hangzhou 310030, China; 3Department of Urology, LMU Klinikum, University Munich, D-81377 Munich, Germany; 4Immunoanalytics Research Group Tissue Control of Immunocytes, Helmholtz Zentrum München, D-81377 Munich, Germany

**Keywords:** cancer stem cells, kidney renal clear cell carcinoma, prognosis, tumor microenvironment, phytochemicals

## Abstract

**Simple Summary:**

Cancer stem cells (CSCs) refer to a group of undifferentiated heterogeneous tumor cells, defined as capable of self-renewal, differentiation, and may be linked to therapeutic resistance and tumor relapse. The development of novel therapeutic strategies to target CSCs and the identification of typical CSC markers are essential to improve therapy efficacy and prevent tumor relapse. Our study identifies CSC markers in renal cell carcinoma (RCC) and explores a potential treatment strategy and the underlying pharmacological mechanisms.

**Abstract:**

The aim of the study was to develop a new therapeutic strategy to target cancer stem cells (CSCs) in clear cell renal cell carcinoma (ccRCC) and to identify typical CSC markers to improve therapy effectiveness. It was found that the corrected-mRNA expression-based stemness index was upregulated in kidney renal clear cell carcinoma (KIRC) tissues compared to non-tumor tissue and increased with higher tumor stage and grade. EZH2 was identified as a CSC marker and prognosis factor for KIRC patients. The expression of EZH2 was associated with several activated tumor-infiltrating immune cells. High expression of EZH2 was enriched in immune-related pathways, low expression was related to several metabolic pathways. Epigallocatechin-3-gallate (EGCG) was identified as the most potent suppressor of EZH2, was able to inhibit viability, migration, and invasion, and to increase the apoptosis rate of ccRCC CSCs. KIF11, VEGF, and MMP2 were identified as predictive EGCG target genes, suggesting a potential mechanism of how EZH2 might regulate invasiveness and migration. The percentages of FoxP3+ Treg cells in the peripheral blood mononuclear cells of ccRCC patients decreased significantly when cultured with spheres pretreated with EGCG plus sunitinib compared to spheres without treatment. Our findings provide new insights into the treatment options of ccRCC based on targeting CSCs.

## 1. Introduction

Renal cell carcinoma (RCC) has become one of the most common genitourinary tumors, of which clear cell renal cell carcinoma (ccRCC) is involved in approximately 70–85% of cases. Over 30% of ccRCC patients face metastasis at the time of initial diagnosis and most of them are not sensitive to chemotherapy and radiation therapy [1,2,3]. At present, cancer stem cells (CSCs) are acknowledged to play an essential role in the metastasis, recurrence, and therapeutic resistance of the tumor through the abilities of self-renewal and differentiation [4,5,6]. It is known that CSCs represent phenotypically and functionally heterogeneous cell populations [7]. The identification of CSC markers as a surrogate for tumor development can indicate tumor metastasis spread and progression.

Enhancer of Zeste Homolog 2 (EZH2), as one of the classical CSC markers, is the key transcriptional regulator involved in histone H3 lysine 27 trimethylation [8]. Previous studies have reported that inhibiting expression can eradicate ovarian CSCs in vitro, blocks disease recurrence, and decreases chemotherapy resistance in acute myeloid leukemia [9,10,11] as well as results in a reduction in the CSC properties and invasion in a number of cancers [12,13,14,15]. In ccRCC, high EZH2 expression was significantly linked to several prognosis values such as shortened overall (*p* < 0.0027) and recurrence free survival (*p* < 0.0001) [16]. More and more CSC markers have been identified, and they play a role in clinical diagnosis and prognosis [17,18,19]. A deeper understanding of the mechanism by which CSC markers mediate and accelerate ccRCC metastasis may help in the development of more efficient treatment strategies. Based on the molecular profiles of normal cells with varying degrees of stemness-associated mRNA expression, a computational algorithm has been developed that calculates a stemness index (mRNAsi) [20], which can be used to describe the similarity between tumor cells and cancer stem cells after controlling the intra-tumor heterogeneity. A higher mRNAsi index was found to be associated with active biological processes in CSCs and greater tumor dedifferentiation, as described by histopathological grades [20,21,22]. So far, a mRNAsi index has not been established for ccRCC.

The tumor microenvironment (TME), as a collaborative interaction between tumor cells and their supporting cells, plays a crucial role in the initiation and progression in ccRCC [23,24]. Tumor-infiltrating immune cells (TICs) in the TME can serve as indicators for the therapeutic effects of immune therapies such as checkpoint inhibition. Moreover, TICs have been found to interact with CSCs to maintain their stemness [25]. Previous studies have found a positive correlation between the CSC markers ALDH and CD3^+^/CD8^+^ cytotoxic T cells and helper T cells in non-small cell lung cancer (NSCLC) [26]. However, no correlation was observed between CD8^+^ TILs (tumor infiltrating lymphocytes) and the CSC marker CD133 in 172 resected NSCLC samples [27]. For the ccRCC patients, several types of TILs were significantly correlated with unfavorable tumor characteristics and a higher rate of recurrence [28,29], but there is less evidence connecting CSC biomarkers with ccRCC TME.

Phytochemicals (PTCs) have obtained attention because of their good safety profile and the ability to target heterogeneous populations of cancer cells as well as CSCs [30,31]. PTCs were found to regulate CSC markers, thereby influencing the biological characteristics of CSCs and several key signaling pathways. For example, epigallocatechin-3-gallate (EGCG), which is an active compound in green tea, shows potential in suppressing cancer stemness and tumorigenicity in several types of cancers [32,33,34,35,36]. Other PTCs showed similar inhibitory effects: wogonin abolished the CSC traits of cutaneous squamous cell carcinoma through downregulating the expression of CSC markers as well as reducing the percentage of CD133^+^ cells; apigenin was found to inhibit the stemness features of triple-negative breast cancer cells by decreasing YAP/TAZ activity [37]. Meanwhile, combining PTCs with other treatment options was observed to further enhance anti-CSC activities [35,38]. Our group found that shikonin, derived from the roots of lithospermum erythrorhizon, combined with cabazitaxel enhanced the anti-tumor effect in prostate CSCs and reversed cabazitaxel resistance by inhibiting the CSC markers ABCG2 and ALDH3A1 [39]. Thus, it is promising to identify typical ccRCC CSC markers and find novel treatment strategies to reverse therapy resistance in cancer patients.

In this study, the effect of EGCG on the expression of the CSC marker EZH2 and the corresponding biological functions were studied using ccRCC lines. Furthermore, the TIC composition was analyzed using the TCGA dataset and deconvolution algorithms and set in relation to the EZH2 expression. Results showed that the expression of EZH2 can be inhibited by EGCG and potentially regulates and affects the prognosis of ccRCC patients as well as TIC composition. Furthermore, the combination of EGCG and sunitinib caused a significant reduction in the FoxP3^+^ Treg cells in the PBMC of patients, indicating potential new treatment options for ccRCC. 

## 2. Materials and Methods

### 2.1. Cell Culture

SKRC-17 (kind gift from J. Vissers, Nijmegen), RCC-53 (derived from a patient with stage IV disease (pT2N1MxG2-3) [19], and RCC-26 (derived from a patient with stage I disease (pT1N0M0G2)) [19,40] cell lines were cultured in RPMI 1640 medium supplemented with 10% fetal calf serum (FCS “Gold Plus”, Bio & Sell GmbH, Feucht, Germany), 2 mM L-glutamine, 1 mM sodium pyruvate, and 1% minimal essential medium (Invitrogen, Life Technologies GmbH, Darmstadt, Germany) at 37 °C in a humidified incubator with 5% CO_2_. CSCs were cultured in DMEM/F12 culture medium, containing 2% B-27 (Invitrogen), 10 ng/mL human recombinant basic fibroblast growth factor (bFGF, Sigma Aldrich Chemie GmbH, Taufkirchen, Germany), and 10 ng/mL epidermal growth factor (EGF, Sigma Aldrich), named as the CSC medium.

### 2.2. Sphere Formation Assay

The SKRC-17, RCC-26, and RCC-53 cells were harvested by StemPro^®^ Accutase^®^ (Life Technologies, Thermo Fisher Scientific, Waltham, MA, USA). Then, cells were filtered through a cell strainer with 40 µM nylon mesh (BD Biosciences, Heidelberg, Germany), and seeded in 75 cm^2^ ultra-low attachment flasks (Corning, New York, NY, USA) with the 10 mL CSC medium. After seven days, photos were taken and the spheres were harvested into a 15 mL tube, centrifuged at 500× *g* for 4 min at room temperature, and dissociated by Accutase for 10 min at 37 °C. After centrifugation, cells were collected and used for the subsequent assays.

### 2.3. Quantitative Reverse Transcription PCR (RT-qPCR)

The total RNA was extracted from cells using a RNeasy Mini-Kit (Qiagen, Hilden, Germany) based on the manufacturer’s instructions. cDNA was synthesized according to the instructions in the kit (Reverse Transcription System, Promega GmbH, Walldorf, Germany). mRNA amplification was performed using the FastStart Essential DNA Green Master kit (Roche, Penzberg, Germany) and LightCycler^®^ 96 (Roche, Basel, Switzerland). The reaction started with 95 °C for 10 min, followed by 40 cycles of denaturation at 95 °C for 10 s, annealing at 60 °C for 10 s, and extension at 72 °C for 10 s. Then, a melting process at 95 °C for 10 s, 65 °C for 60 s, and 97 °C for 1 s was performed. Data were analyzed by the LightCycler^®^ 96 software (1.1 version). The relative expression analysis was carried out by the 2^−ΔΔCt^ method. The transcription level of *GAPDH* and *ACTB* was used as an internal control and all primers are listed in Table 1.

### 2.4. KIRC Data Download

Transcriptome RNA-Seq data of 611 KIRC cases (normal samples, 72 cases; tumor samples, 539 cases) and the corresponding clinical data and stemness indices were downloaded from the TCGA database (https://portal.gdc.cancer.gov/, (accessed on 4 March 2021)).

### 2.5. Survival and Clinical Characteristics Analysis

The R language loaded with package survival and survminer was applied for the survival analysis. A total of 539 tumor samples out of 510 cases had a detailed survival time record, with a time span from 0 to 10 years, which were used for the survival analysis. The Kaplan–Meier method was used to plot the survival curve, and log rank as the statistical significance test; *p* < 0.05 was considered as significant. The analysis of clinical characteristics was performed by R language, the Wilcoxon rank sum or Kruskal–Wallis rank sum test as the significance test, depending on the number of clinical stages for comparison.

### 2.6. Gene Expression in Tumor and Normal Tissue

The mRNA expression level difference and correlation analysis shown in the box plot, heatmap, and the corheatmap were performed by R language software (version 3.6.3, download from https://www.cran.r-project.org (accessed on 17 April 2021)) through the package ggpubr and pheatmap. Immunohistochemistry pictures of potential CSC markers were downloaded from the Human Protein Atlas (www.proteinatlas.org, (accessed on 17 April 2021)) to confirm the protein expression.

### 2.7. Mixed Lymphocyte Tumor Cell Culture (MLTC)

Healthy human donors and patients gave their written informed consent before the experiments were conducted and approved by the Ethics Committee of the Ludwig Maximilians University Munich. Human serum samples were produced from the blood of healthy male donors, which were also approved by the Ethics Committee (Project No. 003/02, Project No. 214/04). Tumor cells were harvested and diluted to 5 × 10^4^ per well with the CTL test medium (Cellular Technology Ltd. Europe, Bonn, Germany) in a 24-well plate. The plate was placed in a sterile tin box in a 37 °C incubator overnight. The next day, the PBMCs were thawed and washed in CTL wash supplemented medium (45 mL RPMI 1640 medium, 5 mL CTL Wash (Cellular Technology Ltd. Europe)) and 50 U/mL benzonase (Novagen Merck Biosciences, Darmstadt, Germany). The PBMC and drugs were added to the tumor cells into a final volume of 1.5 mL per well and incubated for 48 h. Finally, IL-2 in a final concentration of 50 U/mL (Proleukin, Novartis, Basel, Switzerland) was added and the cells were incubated for an additional three days. The workflow for MLTC was conducted by Bio-render (https://biorender.com/, (accessed on 16 May 2022)).

### 2.8. Flow Cytometry

Spheres and adherent tumor cell lines were harvested and stained with the following mouse monoclonal antibodies: EZH2 (clone 11/EZH2 Alexa Fluor^®^ 647, BD Biosciences, Heidelberg, Germany), ALDH1A3 (clone OTI4E8, ORIGENE, Rockville, MD, USA), SALL4 (clone 6E3, Abcam, Cambridge, UK), and ABCG2 (clone 5D3/CD338, APC, BD Biosciences), respectively. As a secondary antibody for the detection of ALDH1A3 and SALL4, a goat F(ab’)2 anti-mouse IgG + IgM (H + L)-FITC antibody (Dianova GmbH, Hamburg, Germany) was used. For intracellular staining, the Cytofix/Cytoperm^TM^ Fixation/Permeabilization Kit from BD Biosciences was used according to the manufacturer’s protocol.

For lymphocyte staining, the following directly conjugated mouse monoclonal antibodies were purchased from BD Biosciences: CD3 (clone UCHT1, FITC), CD4 (clone SK3, PE-Cy7), CD8 (clone RPA-T8, APC), CD25 (clone M-A251, PerCP-Cy5.5), and CD127 (clone hIL-7R-M21, PE), respectively. For Treg analysis, the monoclonal antibody FoxP3 (clone PCH101, eFluor450; eBiosciences, Frankfurt, Germany) was used. Operating steps were as follows: 1–2 × 10^6^ cells were counted and incubated with the LIVE/DEAD™ Fixable Blue Dead Cell Stain (Molecular Probes, Life Technologies, ThermoFisher Scientific, Waltham, MA, USA) for 30 min at room temperature (RT) to stain the dead cells, washed with PBS twice, then stained with the directly labeled monoclonal antibodies for 30 min at 4 °C in the dark, and washed with PBS twice. For intracellular staining with the FoxP3 antibody, the FoxP3/Transcription Factor Staining Buffer Set (eBiosciences) was used and staining was conducted for 60 min at 4 °C in the dark and washed with the Perm buffer twice.

All measurements were accomplished using the LSRII flow cytometer (BD Biosciences). Data analyses were performed by FlowJo software (version 9.9.0; Tree Star Inc., Ashland, OR, USA).

### 2.9. COX Regression Analysis

R language loaded with package survival and survminer were used for univariate and multivariable COX regression. Nine OS-related genes were selected with *p* value < 0.05 by univariate COX regression analysis and eight typical factors of prognosis were analyzed using multivariable COX regression shown in the forest plot.

### 2.10. TICs Profile

The CIBERSORT computational method was applied to estimate the abundance profile of 22 TIC subtypes in the KIRC tumor samples, followed by quality filtering resulting in 539 tumor samples with *p* < 0.05 for display in a bar plot and corheatmap.

### 2.11. Gene Set Enrichment Analysis

KEGG gene sets (c2.cp.kegg.v7.0.symbols.gmt) were downloaded from the Molecular Signatures Database (https://www.gsea-msigdb.org/gsea/msigdb/genesets.jsp, (accessed on 28 January 2021)) as target sets with which GSEA was performed using the software gsea-4.1 downloaded from the Broad Institute. For analysis, gene set permutations were performed 1000 times to obtain a normalized enrichment score, which was used for sorting pathway enrichment, and a NOM *p* < 0.05 and false discovery rate (FDR) of q < 0.06 were considered as significant.

### 2.12. Cell Viability Assay

Cell viability was performed using the CellTiter-Blue Cell Viability Assay (Promega, Mannheim, Germany) according to the manufacturer’s protocol. Cells were seeded into a 96-well microtiter plate (1500 cells/well) and treated with different concentrations of the PTCs EGCG, wogonin, apigenin, and shikonin (all from Selleckchem, Houston, TX, USA). After 24 h and 48 h, the measurement of fluorescence was conducted at a 560(20) excitation/590(10) emission using the FLUOstar OPTIMA microplate reader (BMG LABTECH, Ortenberg, Germany) and the OPTIMA software version 2.0. The half-maximal inhibitory concentration (IC_50_) was assessed by the logit regression model using SPSS version 25.0 (IBM, Armonk, NY, USA).

### 2.13. Apoptosis Assay

The apoptotic assay was assessed by flow cytometry using Annexin V and 7-aminoactinomycin D (both from BD Biosciences). Cells were seeded in a 25 cm^2^ flask at a density of 4 × 10^5^ cells and treated with or without EGCG for five days. Then, the cells were harvested and resuspended in 100 µL Annexin V Binding Buffer with 5 µL APC-Annexin V and 5 µL 7-aminoactinomycin D. After co-incubation for 15 min in the dark at room temperature, 100 µL Annexin V Binding Buffer was added and the cells were measured using a FACSCalibur (Becton Dickinson, San Jose, CA, USA). Data acquisition was conducted using BD CellQuest software (version 4.0.2) and analyzed using FlowJo software (version 9.9.5; Tree Star Inc., Ashland, OR, USA). 

### 2.14. Migration Assay

For cell migration, 24-well culture plates with two small silicone inserts per well were used to establish a cell-free gap of 500 µm (ibidi GmbH, Martinsried, Germany). Cells were seeded into each culture insert at a concentration of 4 × 10^5^ cells/mL in a volume of 70 μL and incubated at 37 °C until a confluent cell monolayer was achieved. Then, the inserts were removed with sterile tweezers and a medium with or without EGCG was added. Pictures were taken under a microscope at different time points (0, 3, 6, 9, 12, 15, and 18 h). The proportion of gap covered by cells at each time was analyzed with the web-based Automated Cellular Analysis System (ACAS, MetaVì Labs, Bottrop, Germany) using FastTrack AI image analysis algorithms.

### 2.15. Invasion Assay

The cell invasion assay was conducted using the Boyden Chamber system. The Transwell inserts (8.0 µm pores; Falcon, Corning, New York, NY, USA) in 24-well plates were coated with growth factor reduced Matrigel Basement Matrix (Corning; 30 µg/100 µL/insert) and incubated at 37 °C for at least 4 h until Matrigel solidification. Then, 30,000 cells were seeded in 250 µL serum-free DMEM medium with and without EGCG onto the Matrigel and the lower chamber was filled with 750 µL DMEM with 10% FCS. After 48 h, the upper surface of the Transwell membrane was gently wiped with a moistened cotton swab to remove Matrigel with the not migrated cells. A total of 4% paraformaldehyde was used to fix the lower surface of the membrane for 5 min and then stained with 1% crystal violet for 1 min. After drying, pictures were taken and cells were counted (three fields per insert, Fiji ImageJ software, download from: https://imagej,net/software/fiji/ (accessed on 18 November 2020)) [41].

### 2.16. Prediction of EGCG Target Genes and Construction of Network

The two-dimensional (2D) and three-dimensional (3D) structure of EGCG was acquired from the PubChem database (https://pubchem.ncbi.nlm.nih.gov/, (accessed on 18 November 2020)). The SDF file of the 3D structure was uploaded to the PharmMapper database (http://www.lilab-ecust.cn/pharmmapper/, (accessed on 28 November 2020)) for potential target gene prediction. The protein–protein interaction (PPI) network was constructed by the STRING database with a confidence score >0.4, followed by the reconstruction with Cytoscape of version 3.6.1 (download from: https://cytoscape.org) [42]. Nodes with a confidence of interactive relationship larger than 0.95 were used to build the network.

## 3. Results

The analysis workflow of our study is shown in Figure 1. A total of 611 KIRC patients from the TCGA database were used to dissect the gene expression profiles, clinicopathological characteristics, and immune-associated molecular classification. Then, the CSC biomarker EZH2 was identified and validated using the KIRC patient data and ccRCC cell lines. The correlation analysis between EZH2 expression and the clinical characteristics, the composition of TICs in KIRC patients, and the signaling pathway analysis were performed. The anti-tumor effect of EGCG against the ccRCC cell lines was evaluated and the potential target genes of EGCG were predicted. The combination therapy of sunitinib with EGCG was tested using coculture experiments and flow cytometry to provide new insights into a new therapeutic option for ccRCC patients.

### 3.1. Corrected-mRNAsi for KIRC

The mRNAsi of ccRCC was downloaded from the TCGA database (KIRC mRNAsi). The mRNAsi expresses the degree of dedifferentiation and similarity between tumor cells and stem cells, which can be considered as a surrogate of CSCs. The mRNAsi as a stemness comprehensive score is derived from the molecular profiles of normal cells with different degrees of stemness based on the OCLR algorithm [20]. Using the mRNAsi, no difference was seen between the normal and tumor tissue, and among the tumor tissues, only high grade tumors had higher mRNAsi (Figure 2A). Previous studies have suggested that tumor “impurities” such as stromal and immune cell types can influence the mRNAsi factor [43,44]. A corrected-mRNAsi was calculated using the ESTIMATE score from previous transcriptomic studies [45]. After correcting by tumor purity, the tumor tissues had a significantly higher stemness index compared to the normal tissue (Figure 2B). The corrected-mRNAsi did significantly increase with the tumor differentiation grade, but also did not correlate to the tumor stage.

### 3.2. Expression Analysis of 19 Potential CSC Marker Genes

Nineteen potential CSC markers for ccRCC based on previous studies were selected [46,47,48]. The mRNA expression values of the tumor and normal tissues of sixteen genes were successfully extracted from the TCGA datasets and are presented as a box plot and heat map (Figure 2C,D). Some of the genes such as *ABCA13*, *CXCR4*, *ABCG2*, *EZH2*, *SALL4*, and *ALDH1A3* were significantly differentially expressed between the ccRCC tissues and normal tissues and showed a higher expression level in sphere cells than in the adherent cells (see later in Figure 3B). The pairs *EZH2* and *DNMT1* (r = 0.57, *p* < 0.0001), *NANOG* and *ABCA13* (r = 0.67, *p* < 0.0001) had the highest correlation among all the pairwise correlations of the sixteen genes, suggesting that they may share some common features or functions (Figure 2E). Univariate COX regression analysis for the survival of ccRCC patients was performed among the markers, determining that seven of them (*ABCG2*, *ALDH1A3*, *ABCB1*, *CD105*, *DAB2IP*, *EZH2*, *SALL4*) had the potential to be a prognosis factor for ccRCC patients (Figure 2F).

### 3.3. Validation of Potential CSC Markers on mRNA and Protein Level

First, we tested the ability of the SKRC-17, RCC-53, and RCC-26 cell lines to generate CSCs using a sphere formation assay over 2–3 passages. Using ultra-low attachment flasks with serum-free medium, CSCs formed non-adherent free-floating spheres, while the differentiated adherent renal cancer cells shrank and died [19,49]. Only SKRC-17 and RCC-53 were capable of generating spheres, as shown in Figure 3A, after 7 days. Next, the mRNA expression levels were analyzed in the SKRC-17 and RCC-53 adherent and sphere cell lines by RT-qPCR (Figure 3B). Four typical CSC markers (EZH2, ABCG2, ALDH1A3, and SALL4) were selected to verify the protein expression. Immunohistochemical images were collected from the Human Protein Atlas (Figure 3C). Flow cytometry was performed to determine the protein expression of the four markers between the adherent and sphere cell lines (Figure 3D). Among them, EZH2 was selected as the most potential ccRCC CSC marker, which was significantly higher expressed in the sphere cell lines than in the adherent cell lines on the mRNA level as well as on the protein level, with the ability to be an OS-related prognostic factor.

### 3.4. Correlation of EZH2 Expression with the Survival and Clinical Characteristics of KIRC Patients

The clinical characteristics of the KIRC patients including survival, stage, grade, and TNM classification were grouped into EZH2 high and low expression according to the median expression level. There was a statistically significant negative correlation between the expression of EZH2 and the prognosis of KIRC patients. The low expression group showed a longer survival than the high expression group (Figure 4A) The expression of EZH2 increased significantly, along with the progression of stage, grade, and TNM stages (Figure 4B). Multivariable Cox regression analysis showed that EZH2 could be an independent prognostic factor to assess the outcomes for KIRC patients (Figure 4C).

### 3.5. Correlation of EZH2 with the Composition of TICs

To explore a potential correlation between CSC features and the KIRC TME, the proportion of tumor-infiltrating immune subsets was determined via CIBERSORT algorithm and correlated to the expression of EZH2. Of the 22 types of immune cell profiles in KIRC samples (Figure 5), nine types of TICs had different frequencies in high versus low expression of EZH2 (Figure 6A–C): follicular helper T cells, regulatory T cells (Tregs), and CD8 T cells were more prevalent in EZH2 high tumors than in the EZH2 low tumors, while B cells, CD4 T memory resting, and M2 macrophages activated dendritic cells as well as resting mast cells were more abundant in EZH2 low tumors.

### 3.6. Functional Enrichment Analysis of EZH2

To further explore the potential association between the immune activity and EZH2, GSEA analysis was implemented for the high expression group compared to the low expression group. As shown in Figure 6D, in the high expression group, immune-related activities such as natural killer cell–mediated cytotoxicity, primary immunodeficiency, and immune signaling pathways such as T cell receptor, p53, and JAK-STAT were mainly enriched. In the low expression group, the genes were enriched, which correlated with the metabolic pathways including pyruvate, propanoate, and butanoate metabolisms.

### 3.7. Influence of Four Different Phytochemicals on EZH2 mRNA Expression

Four phytochemicals (EGCG, shikonin, apigenin, and wogonin) were selected based on the results of previous studies. The SKRC-17 and RCC-53 cell lines were treated with different concentrations of apigenin, EGCG, wogonin, and shikonin, as indicated in the figures, for 24 h and 48 h (Figure 7A). All PTCs inhibited the cell viability in a dose-dependent manner. The half-maximal inhibitory concentration (IC50) was calculated. Then, we treated the SKRC-17 and RCC-53 spheres with different concentrations (0.5 × IC50, 1 × IC50, and 2 × IC50) based on the results from the 48 h treatment to identify the influence on the mRNA expression level of *EZH2* by RT-qPCR (Figure 7B). Transcript levels of *EZH2* in SKRC-17 and RCC-53 CSCs were significantly decreased by shikonin, EGCG, and apigenin, but not significantly by wogonin. EGCG showed the best inhibitory effect of all of the tested PTCs.

### 3.8. EGCG Inhibits Cell Viability and Induces Apoptosis in RCC-53 and SKRC-17 Cell Lines

To further investigate the anti-tumor effect of EGCG, we performed cell viability and apoptosis assays with the RCC-53 and SKRC-17 adherent cell lines and spheres. The results showed a dose-dependent tendency for inhibiting the viability and enhancing apoptosis among the different concentration groups (0.5 × IC50, 1 × IC50, and 2 × IC50) (Figure 8). The SKRC-17 sphere and RCC-53 sphere groups were less sensitive to EGCG than the adherent SKRC-17 and RCC-53 cells.

### 3.9. EGCG Inhibits Migration and Invasion of RCC-53 and SKRC-17 Cell Lines

Migration and invasion are two essential biological characteristics for the metastatic process. Therefore, we evaluated the effect of EGCG on the migration and invasion of SKRC-17, the RCC-53 adherent cell lines, and their corresponding sphere cells. The invasion ability was determined using the Boyden chamber system and cells were treated with different concentrations of EGCG (0.5 × IC50, 1 × IC50, and 2 × IC50) for 48 h. The numbers of invaded cells decreased significantly in the adherent SKRC-17 and RCC-53 cell lines as well as the corresponding spheres in the 2 × IC50 group (Figure 9A). Migration was evaluated using the scratch wound healing assay. After treatment with different concentrations of EGCG (0.25 × IC50, 0.5 × IC50, and 1 × IC50), images were taken at six time points every three hours apart. EGCG in the 1 × IC50 group inhibited the migration of both the adherent cells and sphere cells to the center of the gap, as shown in Figure 9B.

### 3.10. Network of EGCG Target Genes and EZH2

To further uncover the potential pharmacological mechanisms of EGCG inhibiting CSC associated proteins, target genes of EGCG were analyzed in networks. A total of 71 genes were predicted to be direct targets of EGCG, and networks were constructed via the PPI network based on the STRING database. Interactions between the potential 71 target genes and *EZH2* are shown in Figure 10A. Subsequently, a network for EGCG, EZH2 target genes, and *EZH2* was built using the String database and Cytoscape software. As shown in Figure 10B, *EZH2* is connected with the three EGCG potential target genes *VEGFA*, *KIF11*, and *MMP2*.

### 3.11. The Effect of EGCG Combined with Sunitinib on Immune Cells

To determine whether EGCG combined with the first line treatment agent sunitinib had an effect on immune cells, an immunophenotyping assay for the ccRCC patients’ peripheral blood cells (PBMC) was conducted. For this, a co-culture system of CSCs and PBMC was established to study the influence of drug-treated CSCs on different PBMC subpopulations (Figure 11A). The main effect was observed in the Treg cell population. The gating strategy for the Treg cell population is shown in Figure 11B. The CD4^+^, CD25^+^, CD127^low^, FOXP3^+^ T cell subpopulation corresponded to the effector Treg cells. As shown in Figure 11C, the percentage of the FoxP3^+^ Treg subpopulation was significantly decreased in the combination group compared to the group without treatment, the EGCG group, or sunitinib group.

## 4. Discussion

CSCs can evade cell death and can initiate metastasis of the tumor by possessing the capacity of self-renewal and multiple differentiation [50]. Studies have shown that 25–30% of ccRCC patients already had distant metastases at the time of initial diagnosis [51]. So far, angiogenesis inhibition with tyrosine kinase inhibitors (TKIs) combined with the inhibition of immune checkpoint proteins such as PD-1/PDL-1 and CTLA-4 have revolutionized the treatment landscape of ccRCC patients with metastases [52,53]. However, the responses are limited to a subgroup of patients and only approximately 5 to 30% of patients will primarily progress [54]. Therefore, additional therapies or promising additives to the current therapeutic landscape to those non-responders are needed. Recently, it was proposed that renal CSCs isolated from renal cell carcinomas can mediate tumor growth and be the cause for resistance to standard chemo- and radiotherapies, and even tyrosine kinase receptor inhibitors, mTOR inhibitors, VEGF antibodies, cytokines, checkpoint inhibitors, etc. [55,56,57]. Uncovering the molecular mechanisms of metastatic initiation and dynamics of ccRCC CSCs hold promise in improving the management of ccRCC. The key findings of this study can be summarized as follows. (1) a corrected-mRNAsi describing the stemness index of KIRC tumors of the TCGA atlas was significantly high in the KIRC tissue samples compared with normal tissue and positively correlated with the clinical stage and grade. (2) EZH2 was selected as the most promising CSC marker, which was expressed significantly higher in the tumor tissue and cell spheres of the cell lines than in the normal tissue or adherent cell lines, respectively. (3) The expression of EZH2 was statistically correlated with the clinical characteristics, nine types of TICs, immune-related signaling pathways as well as metabolic pathways. (4) Among the four tested PTCs, EGCG suppressed cell viability, migration and invasion, and enhanced apoptosis in adherent and sphere cell lines significantly and showed the highest inhibition of EZH2 expression. *KIF11*, *MMP2*, and *VEGFA* were predicted as target genes interacting with EZH2, and they may play a role and should be tested in future for co-expression. (5) The combination treatment of EGCG with sunitinib significantly decreased the frequency of CD4^+^, CD25^+^, CD127^low^, FOXP3^+^, Treg in the PBMC of the ccRCC patients compared to the group without treatment

Previously, in a pan-cancer stemness index study, the mRNAsi in the KIRC samples was shown to range between 0.2 and 0.5, indicating lower stemness ability than most other tumors such as chromophobe renal cell carcinoma, lymphoma, and uterine sarcoma. After correcting for tumor purity, the corrected-mRNAsi showed a strong prognostic value in KIRC tumors. Not all of the tested ccRCC cell lines possessed stemness capacity (see RCC-26 in Figure 3A), as also described in Gassenmaier et al. [19], and indicates that different ccRCC cells contain various amounts and types of CSC populations. A previous pan-cancer stemness index study identified high expression of EZH2 in ccRCC CSCs [20]. The high prognostic value of EZH2 expression and its association with immune related characteristics observed in this study confirms results from a previous study in which a correlation of high EZH2 expression with poor overall survival was identified in 373 RCC patients, especially in those with advanced disease [58]. The overexpression of EZH2 in RCC associated with lymphocytic infiltration and high CD8^+^ density could navigate immune evasion and the poor patient outcome [16], supporting a treatment strategy aimed at downregulating EZH2. Modulating EZH2 expression or activity has been reported to suppress the phosphorylation of certain receptor tyrosine kinase inhibitors and restore the antitumor effects of sunitinib in models of acquired or intrinsically resistant ccRCC [59]. The proposed mechanism is that downregulated EZH2 expression unleashes pro-inflammatory functions in tumor-infiltrating Tregs cells, which promotes anti-cancer immunity as well as tumor elimination, boosts the recruitment and function of CD8^+^ and CD4^+^ effector T cells, and thus remodels the tumor microenvironment [60]. 

Several EZH2 inhibitors have been developed in recent years, for example, the 2-pyridone compounds GSK126, EPZ005687, and EI1 [61]. The studies have mostly addressed the EZH2-targeted treatments of lymphoma and melanoma or of tumor cell lines. Moreover, other drugs and compounds such as curcumin, omega-3 polyunsaturated fatty acids as well as sorafenib have been reported to be able to downregulate EZH2 and this was critical for their anti-cancer activity, as shown for hepatoma cells, pancreatic tumor cells, and breast cancer cells, respectively [62,63,64,65]. Curcumin, which is another potential anti-tumor phytochemical, was also tested in our group using RCC spheres. An inhibiting effect on spheres was observed with only a slight downregulation of EZH2 (unpublished data). In addition, the inhibition of EZH2 also sensitizes cancer cells to various other anti-cancer drugs such as HDAC inhibitors, imatinib, gemcitabine, paclitaxel, and cisplatin [66,67,68,69,70,71,72,73,74]. In 2020, the Food and Drug Administration (FDA) approved the first EZH2 inhibitor tazemetostat (EPZ-6438) for the treatment of metastatic or locally advanced epithelioid sarcoma [75]. In the kidney, EZH2 plays a major role in maintaining normal kidney function. The overexpression of EZH2 has been demonstrated to be related to acute kidney injury (AKI). It can be speculated that the inhibition of EZH2 overexpression might help prevent the development of pathologies after acute kidney injury. In this regard, the substance zld1039, one of the EZH2 inhibitors, holds promise for the treatment of AKI [76]. Moreover, antibodies against other CSC markers (e.g., CXCR4 antibody, CD105 (endoglin) antibody) have been used in RCC clinical trials (NCT01391130: CXCR4 inhibitor plus sunitinib, NCT01727089: endoglin monoclonal antibody plus bevacizumab, and NCT01806064: endoglin monoclonal antibody plus axitinib). However, all trials failed to improve PFS when combined with TKI agents [77,78,79].

In this study, we hypothesize that EGCG may be a beneficial combination partner for sunitinib for the treatment of ccRCC patients due to its strong inhibitory effect on cancer stem cells (Figure 12). EGCG has been used in in vivo models of various cancers with promising therapeutic effects such as suppressing xenograft tumor growth and invasion, reducing tumor volume and size, inhibiting tumor angiogenesis, and increasing tumor cell apoptosis [31,34]. Our study confirms that EGCG combined with sunitinib showed a significant reduction in FoxP3^+^ Treg cells in the PBMC of patients. Less Treg cells might allow, at least in part, an activation of the endogenous effector CD3^+^CD8^+^ and CD3^+^CD4^+^ T cells in response to the recognition of tumor specific antigens. We propose that the combination of EGCG and sunitinib could be a promising treatment strategy.

This study aimed to identify a CSC marker in ccRCC that could be a potential prognostic marker modulating the microenvironment. Furthermore, we analyzed the mechanisms of the potential phytochemicals’ network pharmacology using bioinformatics, cell biological, and immunological methods. We studied the potential mechanisms of a novel combination treatment strategy for ccRCC patients targeting cancer stem cells and the tumor microenvironment. These findings can help to optimize therapeutic strategies in the future.

## 5. Conclusions

We identified the CSC marker EZH2, which has the potential to be a prognostic and microenvironment-associated factor in ccRCC. Moreover, the novel treatment strategy—EGCG in combination with sunitinib—provides new insights into the treatment options of ccRCC based on targeting CSCs, and the related pharmacological mechanism was further explored in this study.

## Figures and Tables

**Figure 1 cancers-14-04200-f001:**
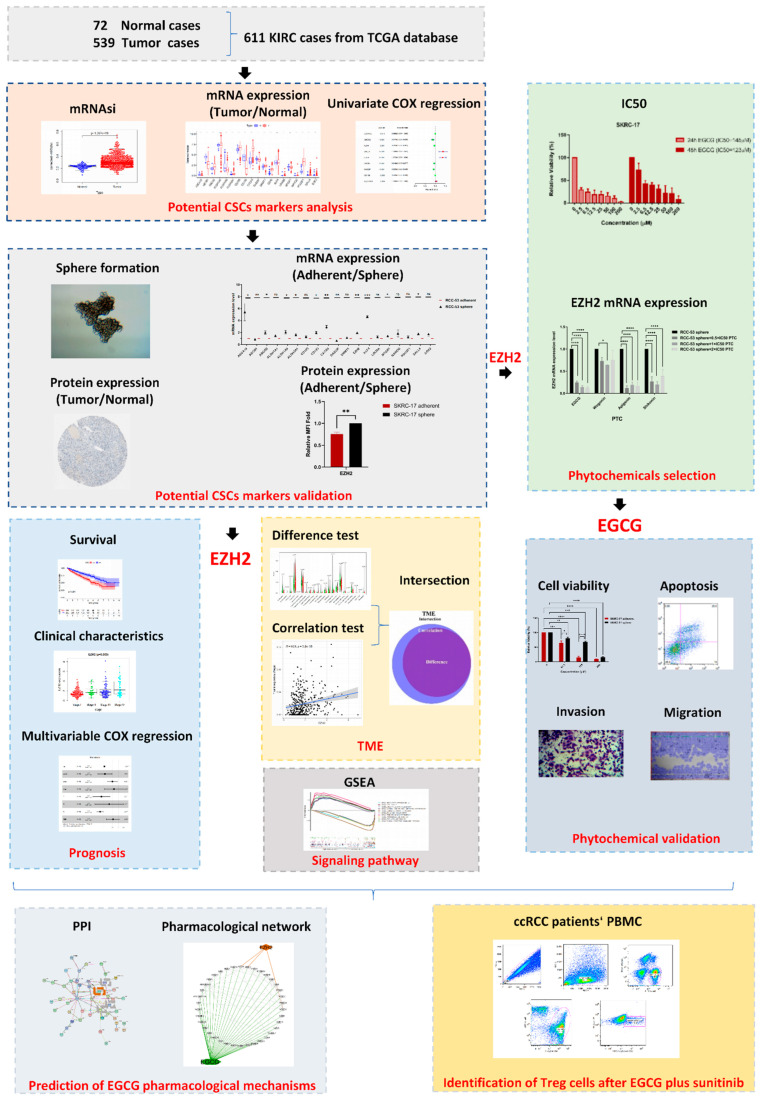
The analysis workflow of this study.

**Figure 2 cancers-14-04200-f002:**
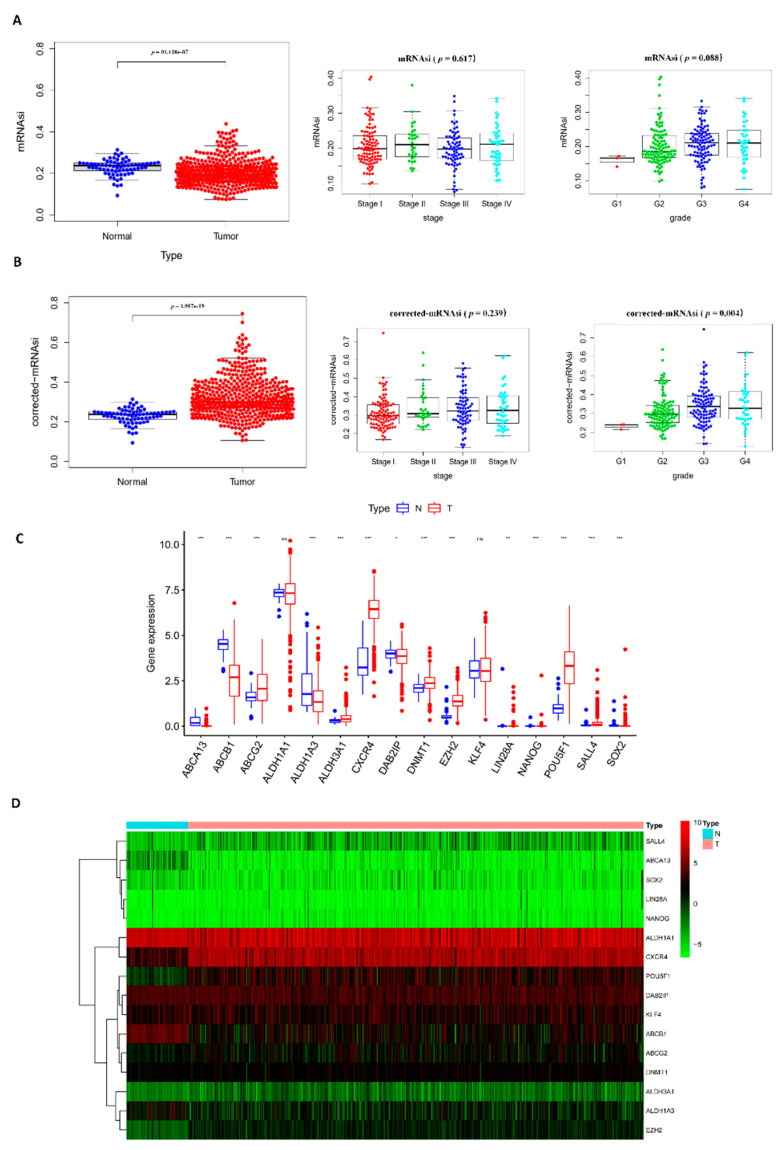
The expression analysis of potential CSC markers in the KIRC tissue and associated normal tissue and univariate COX regression analysis: (**A**) mRNAsi and (**B**) corrected-mRNAsi and association with clinical characteristics in KIRC patients; (**C**) mRNA expression of 19 potential CSC markers, blue indicates normal tissue, and red indicates KIRC tumor tissue; (**D**) heatmap of the expression level of potential CSC markers; (**E**) correlation plot among the expression levels of potential CSC markers based on the Pearson correlation test (*p*  <  0.05 indicates statistical significance); (**F**) univariate COX regression analysis with OS-related CSC markers, significant markers with *p* < 0.005 are listed.

**Figure 3 cancers-14-04200-f003:**
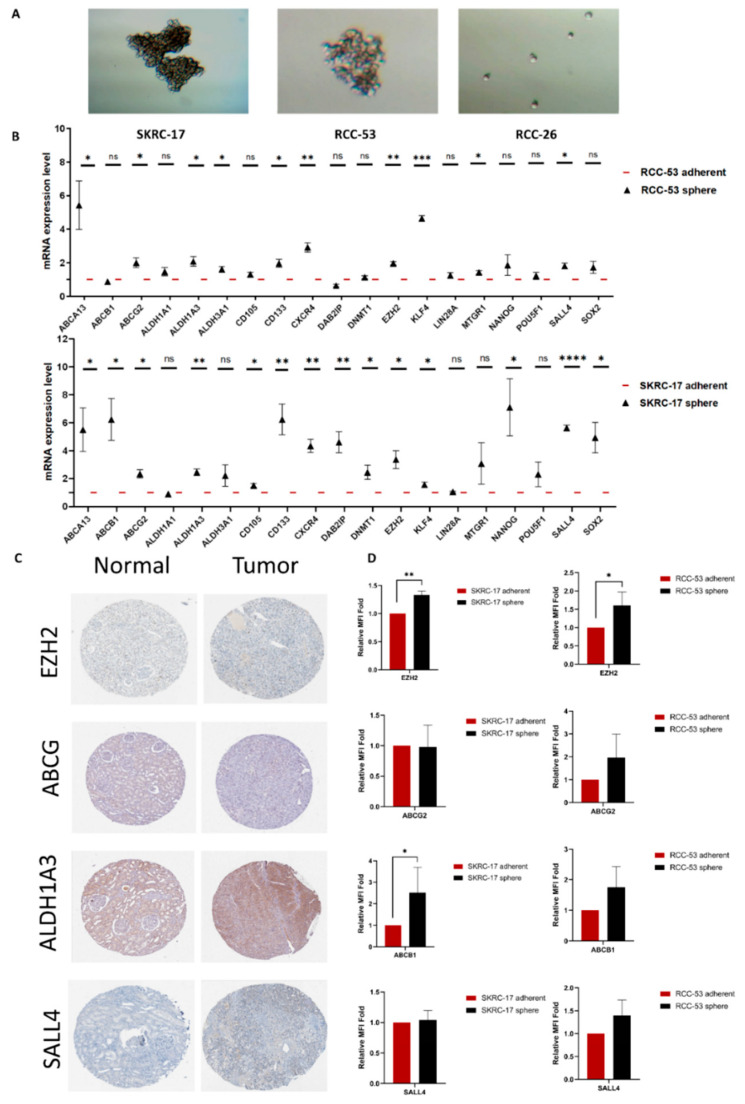
The sphere formation ability of the RCC cell lines and the expression of potential CSC markers on the mRNA and protein levels. (**A**) Sphere formation: SKRC-17 and RCC-53 formed spheres around day 7, RCC-26 did not. Photos were taken by a microscope digital camera at the magnification of 100× (Bresser GmbH DE-46414 Rhede Germany). (**B**) mRNA expression of 19 potential CSC markers in SKRC-17, RCC-53, and their corresponding CSCs analyzed by RT-qPCR analysis (*p*  <  0.05 indicates statistical significance), the expression level observed in spheres was normalized to the corresponding adherent line. * *p* < 0.05, ** *p* < 0.01, *** *p* < 0.001, **** *p* < 0.0001. (**C**) Representative images of IHC staining for the EZH2, ABCG2, ALDH1A3, and SALL4 expression in the KIRC and normal tissue, photos derived from the Human Protein Atlas database. (**D**) Measurement of the protein expression in the adherent and sphere cell lines by flow cytometry (*n* = 3, adherent cell value was set as 1).

**Figure 4 cancers-14-04200-f004:**
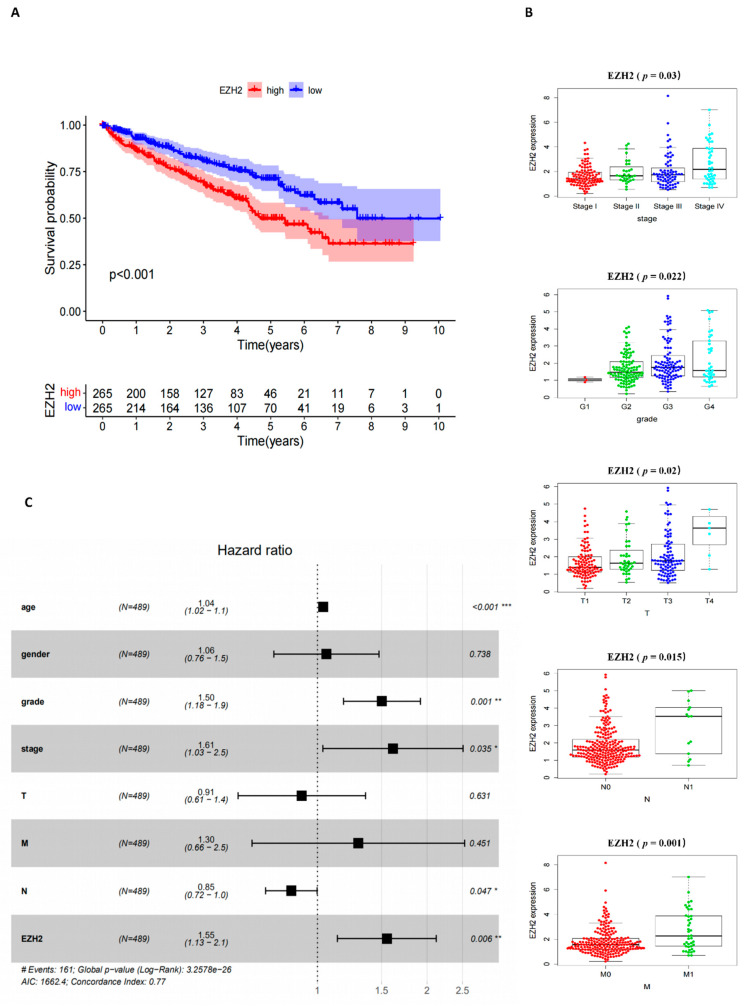
The correlation of EZH2 expression with clinicopathological characteristics: (**A**) Kaplan–Meier survival analysis for the KIRC patients grouped into high or low score in EZH2 expression determined by the comparison with the median, *p* < 0.001 by log-rank test. (**B**) Distribution of EZH2 expression concerning the stage, grade, and T classification, the *p*-values *p* = 0.003, 0.022, and 0.002, respectively, were calculated by the Kruskal–Wallis rank sum test, distribution of scores in M and N classification, the *p*-values *p* = 0.001 and 0.0015, respectively, were calculated by the Wilcoxon rank sum test. (**C**) The multivariate Cox regression analysis of the risk score, age, gender, grade, and TNM stage was used to evaluate the independent prognostic value of EZH2. * *p* < 0.05, ** *p* < 0.01, *** *p* < 0.001.

**Figure 5 cancers-14-04200-f005:**
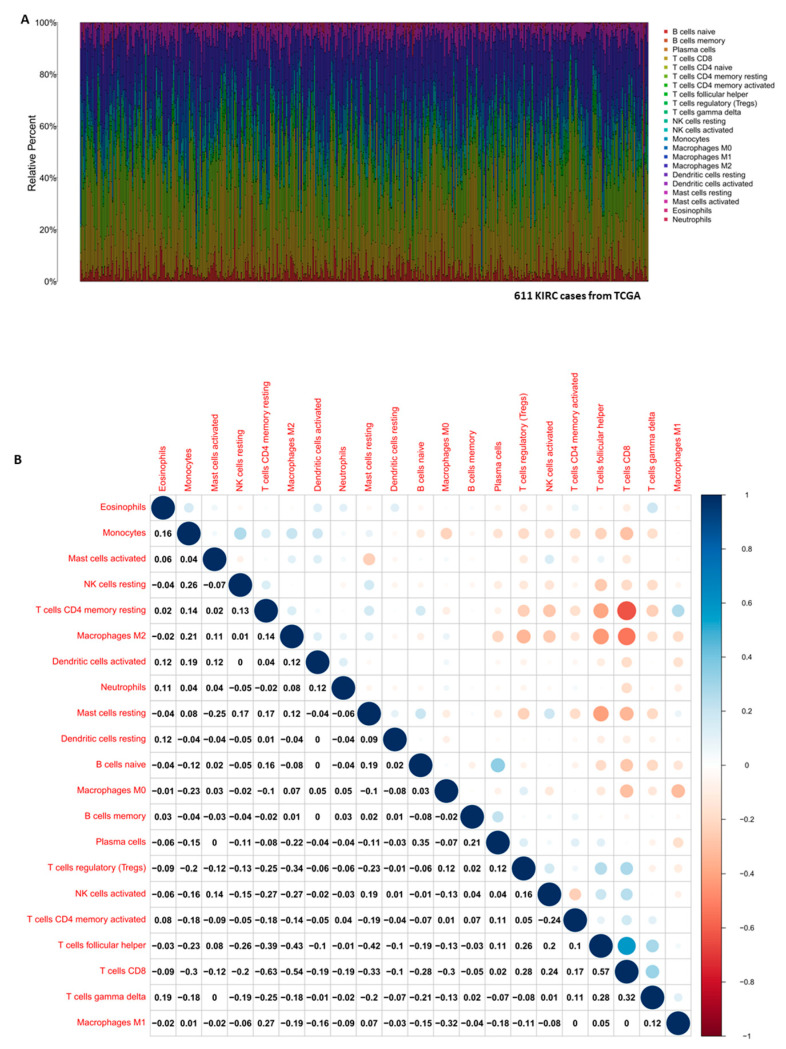
The TIC profile in the KIRC tumor samples and correlation analysis. (**A**) The barplot shows the proportion of 22 types of TICs in the KIRC tumor samples, columns are the 611 KIRC case samples and the different colors are types of TICs. (**B**) The heatmap shows the correlation between 22 TIC populations and the numbers in each tiny box indicate the *p* value of the correlation between two cell populations, the shade of each tiny color box represents the corresponding correlation value between two cell types, and the Pearson coefficient was used for the significance test.

**Figure 6 cancers-14-04200-f006:**
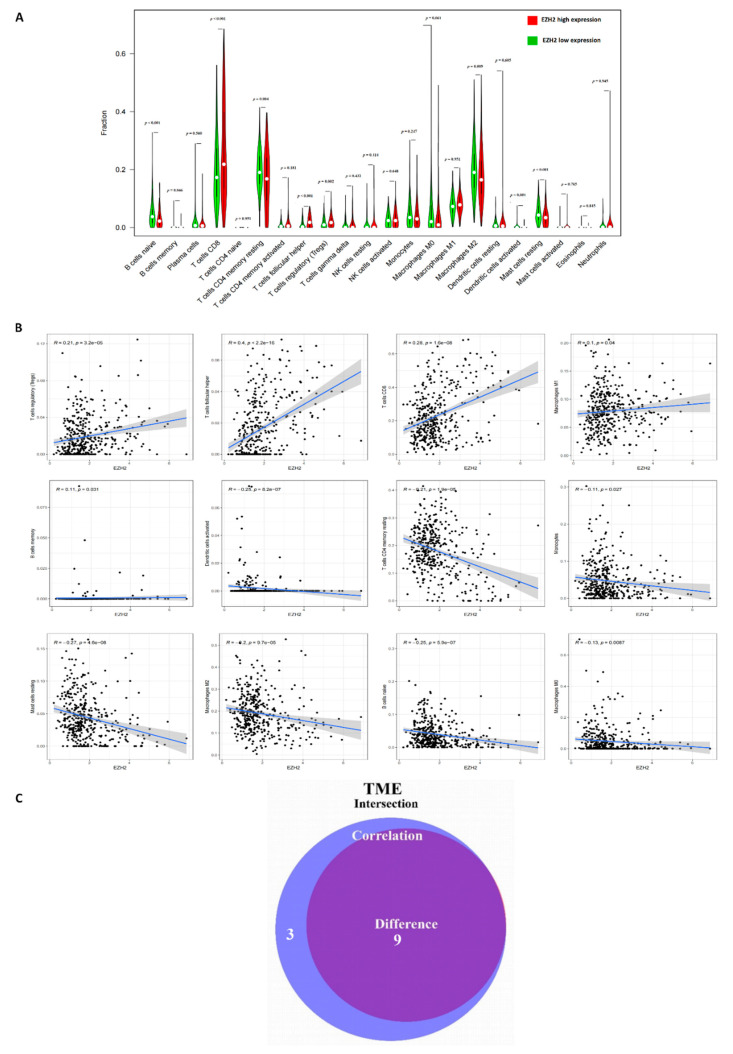
The correlation of the TIC composition and signaling pathway with EZH2 expression. (**A**) Correlation of the TIC population with EZH2 expression, the Violin diagram shows the relationship between the different 22 TIC subpopulations and the KIRC tumor samples with low (green) or high (red) EZH2 expression, relative to the median of the EZH2 expression level, the Wilcoxon rank sum was used for the significance test. (**B**) The scatter plot shows the correlation of 12 TIC subpopulations to the EZH2 expression (*p* < 0.05), the blue line in each plot shows the fitted linear model indicating the proportion tropism of the immune cell along with the EZH2 expression, and the Pearson coefficient was used for the correlation test. (**C**) The Venn plot displays nine TIC subpopulations correlated with the EZH2 expression codetermined by difference and correlation tests displayed in violin and scatter plots, respectively. (**D**) The gene set enrichment analysis for the enriched gene sets with high and low EZH2 expression in the KEGG collection, each line represents one particular gene set with a unique color, upregulated gene sets are located on the left near at the origin of the coordinates, in contrast, the downregulated gene sets are on the right of the *x*-axis. Only gene sets with NOM *p* < 0.05 and FDR q < 0.06 were considered as significant and only some of the leading pathways are displayed in the plot.

**Figure 7 cancers-14-04200-f007:**
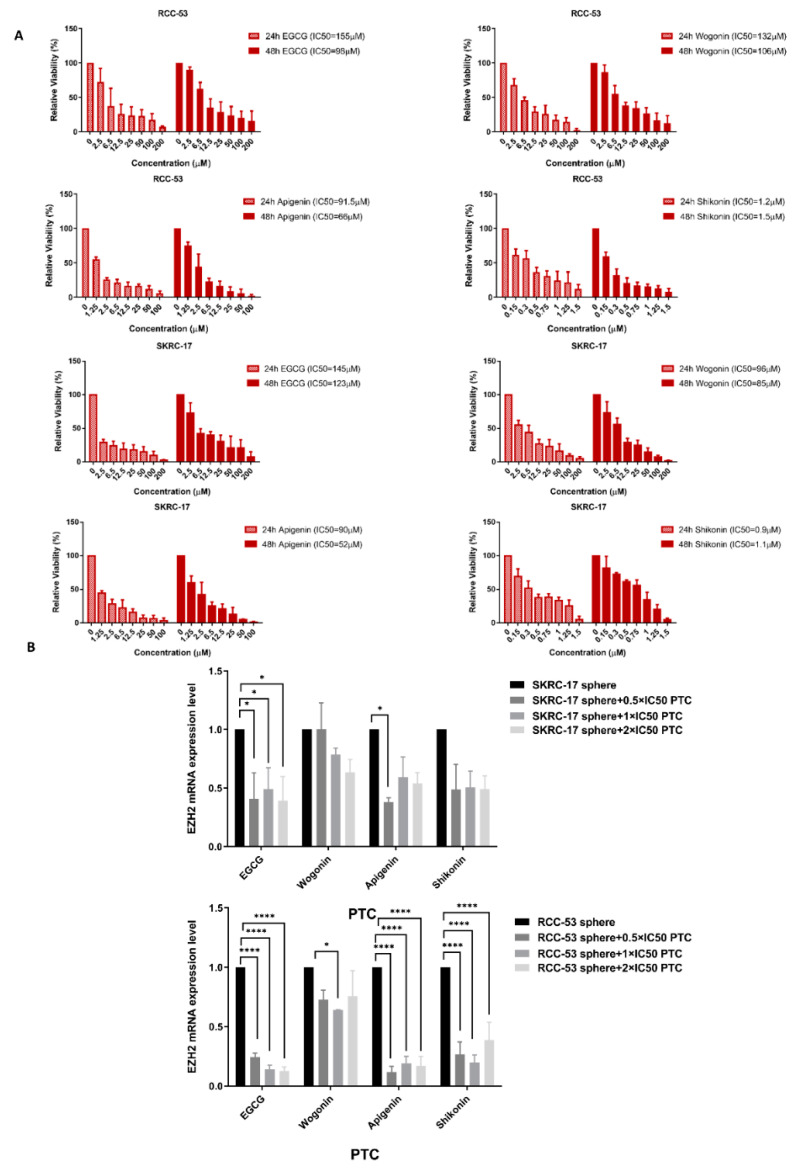
EGCG inhibited the expression of EZH2 in ccRCC CSCs. (**A**) CellTiter-Blue Cell Viability Assay: EGCG, wogonin, apigenin, and shikonin inhibited the viability of SKRC-17, RCC-53 in a dose-dependent manner, shown after 24 and 48 h. The half-maximal inhibitory concentration (IC50) was calculated by the logit regression model. (**B**) RT-qPCR analysis: EGCG, wogonin, apigenin, and shikonin reduced the expression of *EZH2* in the sphere cells, normalized expression levels are displayed (*n* = 3, * *p* < 0.05, **** *p* < 0.0001).

**Figure 8 cancers-14-04200-f008:**
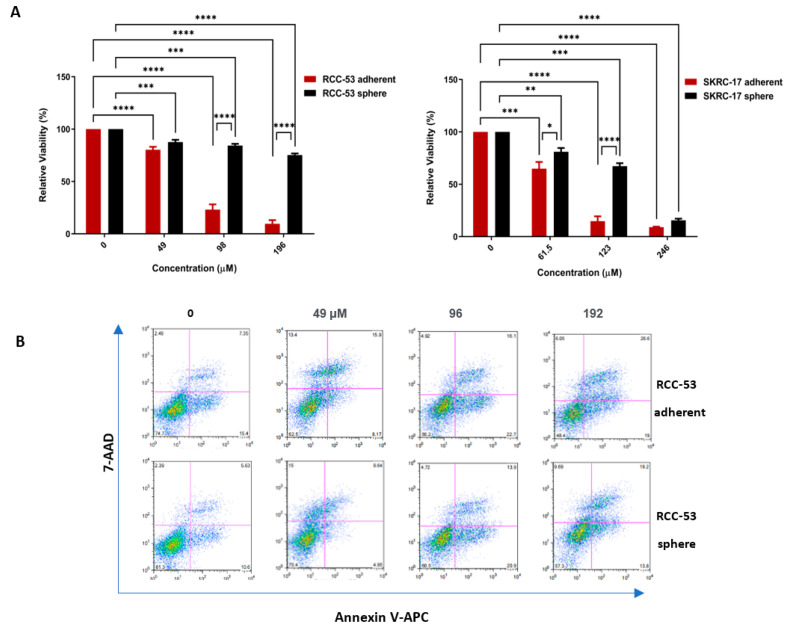
EGCG mediated the anti-tumor effect against SKRC-17, RCC-53, and their corresponding CSCs. (**A**) CellTiter-Blue Cell Viability Assay: EGCG suppressed the cell viability (shown for 48 h). (**B**) EGCG induced apoptosis, the apoptosis rate was calculated by Annexin V and 7-AAD positive staining using flow cytometry (*n* = 3, * *p* < 0.05, ** *p* < 0.01, *** *p* < 0.001, **** *p* < 0.0001).

**Figure 9 cancers-14-04200-f009:**
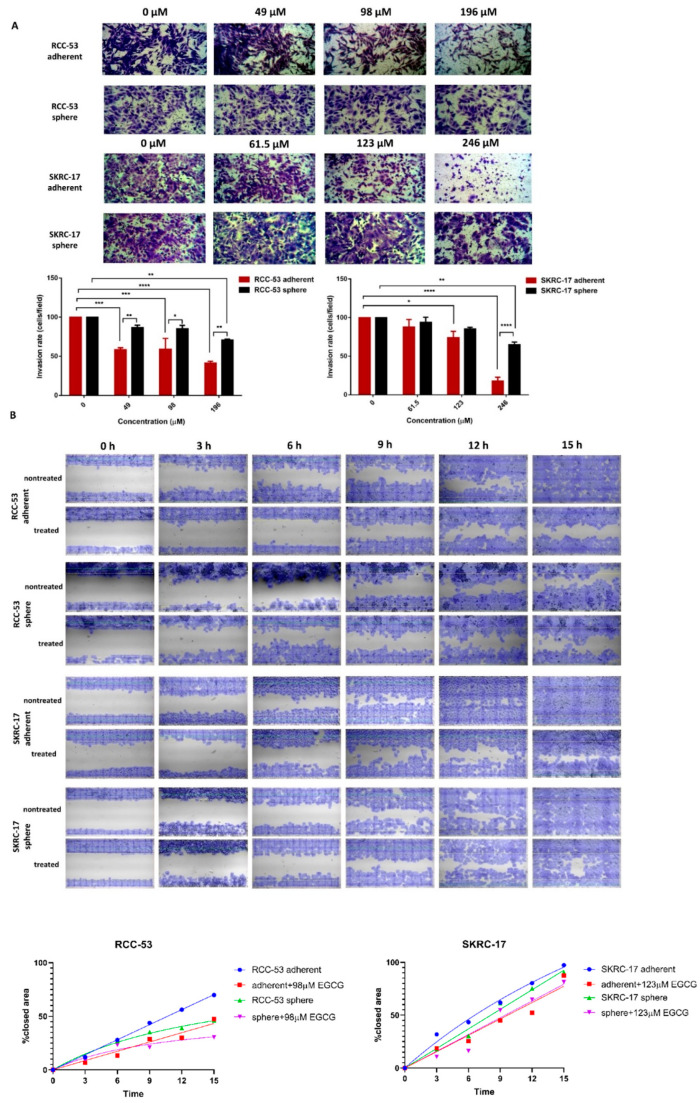
EGCG inhibited invasion and migration of the ccRCC cell lines and their corresponding CSCs. (**A**) For the invasion assay, the Boyden Chamber system was used with Transwell inserts in 24-well plates (8.0 µm pores), coated with growth factor reduced Matrigel Basement Matrix, from every insert a photo was taken at the magnification of 40× and the cells were counted (three fields per insert; Fiji ImageJ software). * *p* < 0.05, ** *p* < 0.01, *** *p* < 0.001, **** *p* < 0.0001. (**B**) For the scratch wound healing, assay plates with a cell-free gap of 500 µm were used (ibidi GmbH), photos were taken under the microscope at different time points at the magnification of 40× (Bresser GmbH DE-46414 Rhede Germany). EGCG in the 1 × IC50 group is shown, and the data were analyzed with the web-based Automated Cellular Analysis System (ACAS, MetaVì Labs) using FastTrack AI image analysis algorithms.

**Figure 10 cancers-14-04200-f010:**
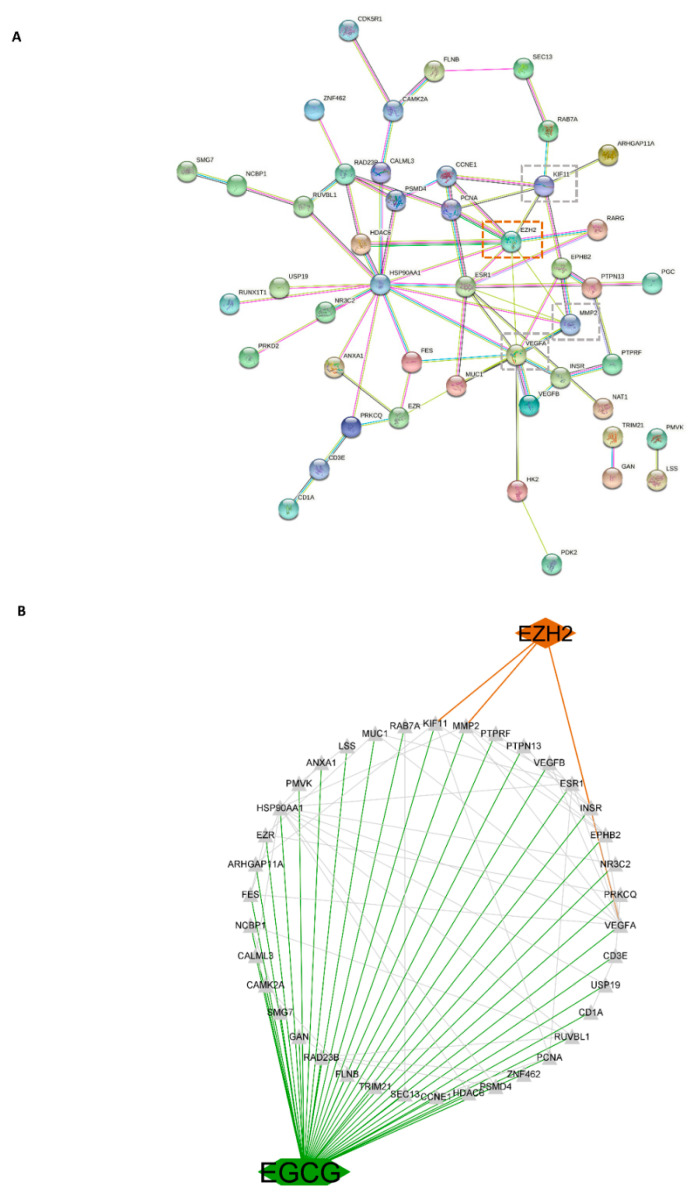
The interaction network based on the potential targets of EGCG. (**A**) The PPI network of potential target genes of EGCG and EZH2; the edges represent the predicted functional associations, green line: the neighborhood evidence, blue line: co-occurrence evidence, purple line: experimental evidence, yellow line: text mining evidence, black line: co-expression evidence, dotted lines: the potential target genes of EGCG and EZH2. (**B**) The network of EGCG, EZH2 target genes, and EZH2, green lines represent the interaction between EGCG and its target genes, red lines represent interaction between EZH2 and target genes, black lines represent interaction among target genes.

**Figure 11 cancers-14-04200-f011:**
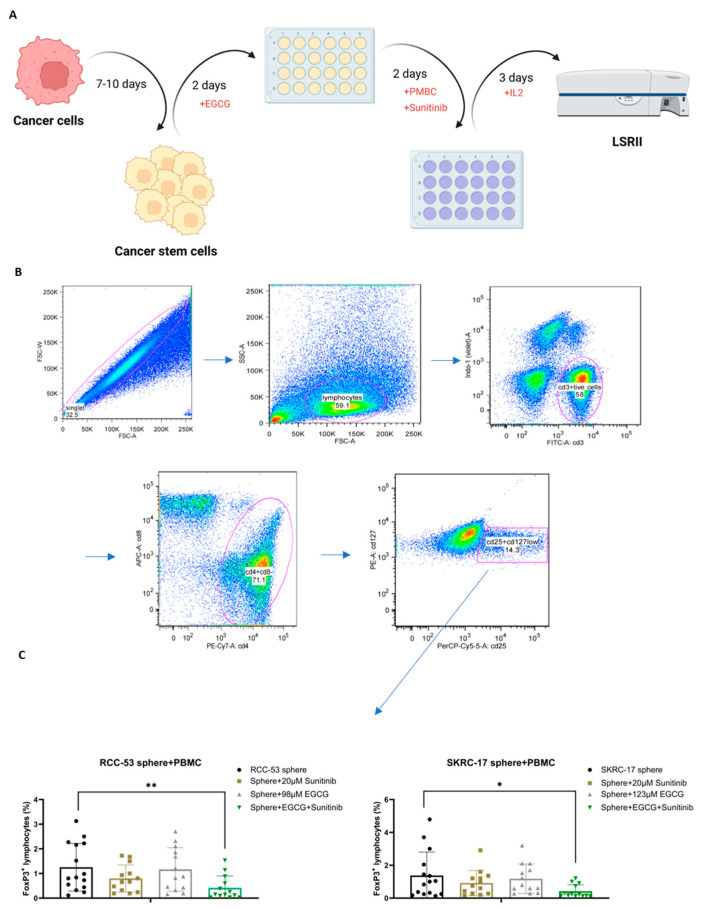
The Treg cells were significantly decreased by the combination treatment EGCG plus sunitinib in co-culture with CSCs. (**A**) Workflow of the MLTC system. (**B**) Representative plots showing the gating strategy for Treg analysis. (**C**) Percentages of FoxP3^+^CD25^+^CD127^low^ lymphocytes among the CD4^+^ T cells (* *p* < 0.05, ** *p* < 0.01).

**Figure 12 cancers-14-04200-f012:**
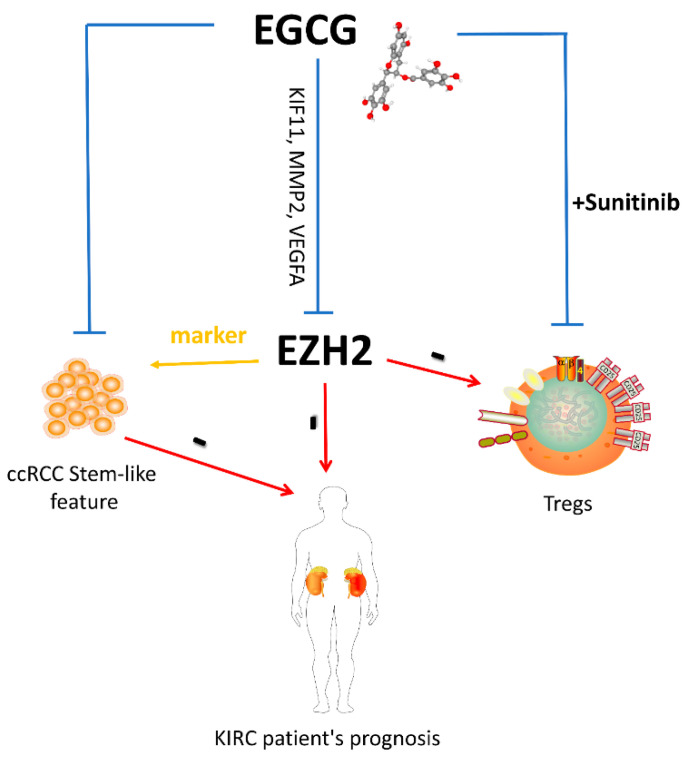
The proposed model for the correlation among EZH2, cancer stem cells, and EGCG for the KIRC patients.

**Table 1 cancers-14-04200-t001:** The primers used for RT-qPCR.

Transcript	Primer	Sequence (5′-3′)	Product Size (bp)
*GAPDH*	GAPDH-F	CATGGGTGTGAACCATGA	104
	GAPDH-R	TGTCATGGATGACCTTGG	
*ACTB*	ACTB-F	CTGCCCTGAGGCACTC	197
	ACTB-R	GTGCCAGGGCAGTGAT	
*ABCA13*	ABCA13-f	AGGAGTGTGAGGCTCTTTGC	207
	ABCA13-r	TCAGGTGCTGTCCCTTGAAC	
*ABCB1*	ABCB1-f	GGAGGCCAACATACATGCCT	205
	ABCB1-r	CAGGGCTTCTTGGACAACCT	
*ABCG2*	ABCG2-f	CATCAACTTTCCGGGGGTGA	266
	ABCG2-r	CACTGGTTGGTCGTCAGGAA	
*ALDH1A1*	ALDH1A1-f	TGTTAGCTGATGCCGACTTG	154
	ALDH1A1-r	TTCTTAGCCCGCTCAACACT	
*ALDH1A3*	ALDH1A3-f	GAGGAGATTTTCGGGCCAGT	186
	ALDH1A3-r	GAGGGCGTTGTAGCAGTTGA	
*ALDH3A1*	ALDH1A3-f	GCAGACCTGCACAAGAATGA	186
	ALDH1A3-r	TGTAGAGCTCGTCCTGCTGA	
*CD105*	ENG-f	TCACCACAGCGGAAAAAGGT	141
	ENG-r	GGACACTCTGACCTGCACAA	
*CD133*	PROM1-f	TTGCGGTAAAACTGGCTAAG	155
	PROM1-r	TGGGCTTGTCATAACAGGAT	
*CXCR4*	CXCR4-f	TGGGTGGTTGTGTTCCAGTTT	80
	CXCR4-r	ATGCAATAGCAGGACAGGATG	
*DAB2IP*	DAB2IP-f	TGTCGCCCTCACTCTTCAAC	225
	DAB2IP-r	CGGCTGTATTGGAGAGGGTC	
*DNMT1*	DNMT1-f	GGCAGACCATCAGGCATTCT	220
	DNMT1-r	ACCATGTCCTTGCAGGCTTT	
*EZH2*	hEZH2-F	AGGACGGCTCCTCTAACCAT	179
	EZH2-R	CTTGGTGTTGCACTGTGCTT	
*KLF4*	KLF4-f	TCCCATCTTTCTCCACGTTC	239
	KLF4-r	GGTCTCTCTCCGAGGTAGGG	
*LIN28A*	LIN28A-f	TTCGGCTTCCTGTCCATGAC	124
	LIN28A-r	CCACTGCCTCACCCTCCTT	
*MTGR1*	MTGR1-f	CCTCCTACCCTGAATGGTGC	214
	MTGR1-r	GTGCAAGAACAAGAGTCCGC	
*NANOG*	NANOG-f	TGTGTTCTCTTCCACCCAGC	205
	NANOG-r	CTTCTGCGTCACACCATTGC	
*POU5F1*	POU5F1-f	CCCTGGGGGTTCTATTTGGG	231
	POU5F1-r	TCTCCAGGTTGCCTCTCACT	
*SALL4*	SALL4-f	GCTCTGTTAGGTACGGACGG	96
	SALL4-r	CTGGTTCCACACAACAGGGT	
*SOX2*	SOX-2-f	CATCACCCACAGCAAATGAC	258
	SOX-2-r	GCAAACTTCCTGCAAAGCTC	

## Data Availability

Data are contained within the article from the corresponding author on reasonable request.

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
