# Peer review of "Identification of EZH2 as Cancer Stem Cell Marker in Clear Cell Renal Cell Carcinoma and the Anti-Tumor Effect of Epigallocatechin-3-Gallate (EGCG)"

_cancers, 2022, doi:10.3390/cancers14174200_

Round 1
Reviewer 1 Report
Gist/summary:
The authors study, the effect of EGCG on the expression of the Cancer stem cell (CSC) marker, viz. EZH2. They employed ccRCC lines and TCGA database and checked the expression. As a result, they showed that the expression of EZH2 can be inhibited by EGCG in lieu fo regulatin. They plotted the survival abakyses and in combination of well known drug sunitinib, they showed a reduction of FoxP3+ Treg cells
Strengths:
The analysis is robust with a good number of wet lab experiments with algorithmic components. The figures are plotted well. Th epictorial methodology is very nice
Weaknesses: Starting with in silico and ending with in vitro validation would have been nicer, IMHO
What is the software used for gene expression abundance in R?
While doing pharmacological studies, why didn't authors cherry pick the drug and check for docking with the drug? Although the ECGC target genes were done, I thought the other way round, first employing the target gene models and finding those coexpressed genes woul dbe a good idea
How would the authors justify small number of control datasets in KIRC/TIRC database?
The STRING interaction map is nice. Instead of lines, the word edges may be used
Minor but essential edits:
In introduction,
Pl remove "state" after resistance
Pl use plural indicatorS for " TME can serve as indicator .."
In Abstract, pl add "a" to CSC
In Materials and Methods, pl add "a" before 37C incubator
Pl add "the" before manufacturer's
In Results, pl add "the" before TCGA database
Pl correct therapeutical to "therapeutic"
database is one word
Pl correct differently to "differentially"
Pl correct to "T Cells" ( pluralise)
Please add "the" in "We propsoe THE combination
Pl correct "bioinformatical" to "bioinformatics"
PL add a few lines about conclusions
Scores on a scale of 0-5 with 5 being the best
Language: 4
Novelty: 4
Scope/Relevance: 4
Brevit: 3.5
Author Response
Responses to Reviewer 1:
What is the software used for gene expression abundance in R?
Thank you for pointing out this problem in our manuscript. Gene expression abundance (Figure 2 C, D) were performed by R language software(version 3.6.3)by packages-ggpubr and pheatmap. These details are added to the Method Part- “Gene expression in tumor and normal tissue” in the manuscript (Page 5).
While doing pharmacological studies, why didn't authors cherry pick the drug and check for docking with the drug? Although the ECGC target genes were done, I thought the other way round, first employing the target gene models and finding those coexpressed genes would be a good idea.
Thanks for your brilliant suggestion. It would be a good idea to improve the design of the study. Molecular docking of the EGCG was tentatively organized during the experiment duration, but due to the limited knowledge of the pharmacology and the relative prediction tools, we did not succeed for this part, more details of this part can be hopefully explored in the future.
How would the authors justify small number of control datasets in KIRC/TIRC database?
First, the KIRC data were controlled by TCGA database itself, based on a quality control protocol to ensure that molecular analytes (i.e. DNA and RNA) extracted from tissue samples are of consistent and high quality. Then, we distinguished and calculated the numbers of tumor and normal tissue by the software-Perl based on the sample’s annotation, meanwhile only the qualified samples, which contains enough information would be left and proceeded in the next part. Finally, in terms of the small number of control datasets in the database, the control samples selected for further analysis would be marked as normal sample and two groups were distinguished by the case numbers to process to the next analysis.
The STRING interaction map is nice. Instead of lines, the word edges may be used.
Thank you for the above suggestion. The explanation of the different colours of edges were described in the legend (Figure 10 A) - “the edges represent the predicted functional associations, green line: the neighbourhood evidence, blue line: co-occurrence evidence, purple line: experimental evidence, yellow line: text mining evidence, black line: co-expression evidence, dotted lines: the potential target genes of EGCG and EZH2”.
Minor but essential edits:
Thank you for pointing out this problem in our manuscript. According to the revised content, we corrected these mistakes and details, see below:
In introduction,
Pl remove "state" after resistance
Corrected on page 2:
cancer stem cells (CSCs) are acknowledged to play an essential role in metastasis, recurrence, and therapeutic resistance state of the tumor by the abilities of self-renewal and differentiation
into
cancer stem cells (CSCs) are acknowledged to play an essential role in metastasis, recurrence, and therapeutic resistance of the tumor by the abilities of self-renewal and differentiation.
Pl use plural indicators for " TME can serve as indicator .."
Corrected on page 2:
Tumor-infiltrating immune cells (TICs) in the TME can serve as indicator for the therapeutic effects of immune therapies like checkpoint inhibition.
Into
Tumor-infiltrating immune cells (TICs) in the TME can serve as indicators for the therapeutic effects of immune therapies like checkpoint inhibition.
In Abstract, pl add "a" to CSC
Corrected on page 1:
EZH2 was identified as CSC marker and prognosis factor for KIRC patients.
Into
EZH2 was identified as a CSC marker and prognosis factor for KIRC patients.
In Materials and Methods, pl add "a" before 37C incubator
Corrected on page 3:
at 37 °C in humidified incubator with 5% CO2
into
at 37 °C in a humidified incubator with 5% CO2
Pl add "the" before manufacturer's
Corrected on page 5:
Cytofix/CytopermTM Fixation/Permeabilization Kit from BD Biosciences was used according to manufacturer’s protocol.
Into
Cytofix/CytopermTM Fixation/Permeabilization Kit from BD Biosciences was used according to the manufacturer’s protocol.
In Results, pl add "the" before TCGA database
Corrected on page 7:
The analysis workflow of our study is shown in Figure 1. A total of 611 KIRC patients from TCGA database were used to dissect gene expression profiles
Into
The analysis workflow of our study is shown in Figure 1. A total of 611 KIRC patients from the TCGA database were used to dissect gene expression profiles.
Pl correct therapeutical to "therapeutic"
Corrected on page 7:
The combination therapy of sunitinib with EGCG was tested using coculture experiments and flow cytometry to provide new insights into a new therapeutical option for ccRCC patients
Into
The combination therapy of sunitinib with EGCG was tested using coculture experiments and flow cytometry to provide new insights into a new therapeutic option for ccRCC patients
database is one word
corrected on page 9
The mRNAsi of ccRCC was downloaded from the TCGA data base (KIRC mRNAsi)
Into
The mRNAsi of ccRCC was downloaded from the TCGA database (KIRC mRNAsi).
Pl correct differently to "differentially"
Corrected on page 9:
such as ABCA13, CXCR4, ABCG2, EZH2, SALL4, and ALDH1A3 were significantly differently expressed between ccRCC tissues and normal tissues and showed a higher expression level in sphere cells than in adherent cells
Into
such as ABCA13, CXCR4, ABCG2, EZH2, SALL4, and ALDH1A3 were significantly differentially expressed between ccRCC tissues and normal tissues and showed a higher expression level in sphere cells than in adherent cells
Pl correct to "T Cells" ( pluralise)
Corrected on page 31:
percentages of FoxP3+CD25+Cd127low lymphocytes among CD4+ T cell.
Into
percentages of FoxP3+CD25+CD127low lymphocytes among CD4+ T cells.
Please add "the" in "We propsoe THE combination
Corrected on page 32:
We propose that combination of EGCG and sunitinib could be a promising treatment strategy.
Into
We propose that the combination of EGCG and sunitinib could be a promising treatment strategy.
Pl correct "bioinformatical" to "bioinformatics"
Corrected on page 33:
Furthermore, we analyzed the mechanisms of potential phytochemicals’ network pharmacology using bioinformatical, cell biological, and immunological methods.
Into
Furthermore, we analyzed the mechanisms of potential phytochemicals’ network pharmacology using bioinformatics, cell biological, and immunological methods.
PL add a few lines about conclusions
We changed conclusions on page 33:
“We identified the CSC marker EZH2, which has the potential to be a prognostic and microenvironment-associated factor in ccRCC, and further explored a treatment strategy and the related pharmacological mechanism”
Into
“We identified the CSC marker EZH2, which has the potential to be a prognostic and microenvironment-associated factor in ccRCC. Moreover, the novel treatment strategy-EGCG in combination with sunitinib provides a new insight into treatment options of ccRCC based on targeting CSCs, and the related pharmacological mechanism was further explored in this study.”
Reviewer 2 Report
cancers-1846538
Title: Identification of EZH2 as cancer stem cell marker in clear cell renal cell carcinoma and the anti-tumor effect of epigallocatechin-3-gallate (EGCG)
Major revision
In this paper, authors investigated to develop a new therapeutic strategy to target cancer stem cells (CSCs) in clear cell renal cell carcinoma (ccRCC) and to identify typical CSC markers to improve therapy effectiveness. However, this paper is not enough to clearly explaining these mechanisms.
1. As you know, EZH2 expression accurately predicts risk of RCC death beyond existing clinicopathologic models, However, further studies are required to incorporate molecular biomarkers into cancer cell microenvironments.
2. Recently, several highly selective small molecule inhibitors against EZH2, such as GSK126, EPZ005687, EI1, and EPZ-6438, have been developed. In addition to specific EZH2 inhibitors, several other drugs and compounds have been reported to be able to downregulate EZH2, and the downregulation of EZH2 is critical for their anti-cancer activity. These include curcumin, omega-3 polyunsaturated fatty acids, and sorafenib. Moreover, inhibition of EZH2 also sensitizes cancer cells to various other anti-cancer drugs, such as HDAC inhibitors, imatinib, gemcitabine, paclitaxel, and cisplatin. What is different these previous studies ?
3. What is the major function of EGCG on the targeting EZH2 overexpression tumor ?
4. Which biomarkers are closely associated with the RCC cells migration and invasion?
5. Authors indicated that SKRC-17 sphere and RCC-53 sphere groups were less sensitive to EGCG than the adherent SKRC-17 and RCC-53 cells. Why ?
6. Authors measured the effect of EGCG on migration and invasion of SKRC[1]17, RCC-53 adherent cell lines. However, the expression of MMPs and TIMPs should be compared.
Author Response
Responses to Reviewer 2:
- As you know, EZH2 expression accurately predicts risk of RCC death beyond existing clinicopathologic models, However, further studies are required to incorporate molecular biomarkers into cancer cell microenvironments.
Thank you for your comment. We carefully considered your suggestion that “further studies are required to incorporate molecular biomarkers into cancer cell microenvironments”. Therefore, we deconvoluted the EZH2-related microenvironments in KIRC via bioinformatic analysis. Nine types of tumor-infiltrating immune cells had different frequencies in tumors with high or low expression of EZH2. Follicular helper T cells, regulatory T cells (Tregs), CD8 T cells were more prevalent in EZH2-high tumors than in EZH2-low tumors, while B cells, CD4 T memory resting, M2 macrophages activated dendritic cells as well as resting mast cells were more abundant in EZH2-low tumors.
- Recently, several highly selective small molecule inhibitors against EZH2, such as GSK126, EPZ005687, EI1, and EPZ-6438, have been developed. In addition to specific EZH2 inhibitors, several other drugs and compounds have been reported to be able to downregulate EZH2, and the downregulation of EZH2 is critical for their anti-cancer activity. These include curcumin, omega-3 polyunsaturated fatty acids, and sorafenib. Moreover, inhibition of EZH2 also sensitizes cancer cells to various other anti-cancer drugs, such as HDAC inhibitors, imatinib, gemcitabine, paclitaxel, and cisplatin. What is different these previous studies?
EZH2, the catalytic subunit of the Polycomb repressive complex 2 (PRC2), catalyzes the methylation of lysine 27 of histone H3 (H3K27) up to its trimethylated form (H3K27me). EZH2 inhibitors such as GSK126, EPZ005687, EI1 and EPZ-6438 belong to the 2-pyridone compounds with a central heteroaromatic bicyclic or monocyclic core identified as potent and selective catalytic inhibitors in both EZH2 wild-type and EZH2 mutant cells [1]. The 2-pyridone occupies partially the site for the co-substrate S-adenosyl-L-methionine- (SAM) in the binding pocket of the enzyme, accounting for the SAM-competitive mechanism of action displayed by all the 2-pyridone inhibitors. Most studies have focused on EZH2-targeted treatments in lymphoma and melanoma, or on cell lines. So far, Food and Drug Administration (FDA)-approved first EZH2 inhibitor is tazemetostat (EPZ-6438) for the treatment of metastatic or locally advanced epithelioid sarcoma in 2020. Few data exist for RCC and related cell lines. We also identified EZH2 as a new marker for RCC stem cells, which are discussed as responsible for drug resistance. EZH2-targeted therapy can be one of the treatment strategies that can potentially improve patient survival more efficiently. We were able to show that the combination therapy EGCG plus sunitinib (standard therapeutic drug in metastatic RCC) was also able to inhibit the expression of EZH2 on CSCs.
The drugs and compounds like curcumin, omega-3 polyunsaturated fatty acids, and sorafenib had been reported to be able to downregulate EZH2 and this activity was critical for their anti-cancer activity in hepatoma cells [2], pancreatic tumor cells[3] , breast cancer cells [4,5], respectively. We focused on RCC CSCs and we used patients’ PBMC samples to characterize the potential of the combination therapy on the immunological level. Moreover, curcumin was also tested in our group using RCC spheres. An inhibiting effect on spheres was observed but only a slight downregulation of EZH2 (Sarah Hubschneider, unpublished data).
Inhibition of EZH2 also sensitizes cancer cells to various other anti-cancer drugs, such as HDAC inhibitors, imatinib, gemcitabine, paclitaxel, and cisplatin [2,6-14]. So far, there is no EZH2-targeted research in RCC, and these anti-cancer drugs are not the first-line drugs for RCC either. Therefore, our study fills the gap in this area and proposes a potentially highly effective treatment strategy for RCC.
We appreciated your comments and added more information about EZH2 inhibition in the discussion as following (page 32):
Several EZH2 inhibitors have been developed in the recent years, for example the 2-pyridone compounds GSK126, EPZ005687, and EI1 [61]. The studies mostly address EZH2-targeted treatments of lymphoma and melanoma or of tumor cell lines. Moreover, other drugs and compounds like curcumin, omega-3 polyunsaturated fatty acids as well as sorafenib have been reported to be able to downregulate EZH2 and this was critical for their anti-cancer activity, shown for hepatoma cells, pancreatic tumor cells, and breast cancer cells [62-65]. Curcumin, which is another potential anti-tumor phytochemical, was also tested in our group using RCC spheres. An inhibiting effect on spheres was observed with only a slight downregulation of EZH2 (unpublished data). In addition, inhibition of EZH2 also sensitizes cancer cells to various other anti-cancer drugs, such as HDAC inhibitors, imatinib, gemcitabine, paclitaxel, and cisplatin [66-74]. In 2020, the Food and Drug Administration (FDA)-approved the first EZH2 inhibitor tazemetostat (EPZ-6438) for the treatment of metastatic or locally advanced epithelioid sarcoma [75]. In the kidney, EZH2 plays a major role in maintaining normal kidney function. Overexpression of EZH2 has been demonstrated to be related to kidney injury (AKI). It can be speculated, that inhibition of EZH2 overexpression might help prevent the development of pathologies after acute kidney injury. In this regard, the substance zld1039, one of the EZH2 inhibitors, holds promise for the treatment of AKI [76]. Moreover, antibodies against other CSC markers (e. g. CXCR4 antibody, CD105 (endoglin) antibody) have been used in RCC clinical trials.
- What is the major function of EGCG on the targeting EZH2 overexpression tumor?
The aim of this study was to inhibit CSCs in RCC in order to reverse drug resistance and improve the efficacy of therapy. We identified EZH2 as CSC marker. EZH2 was found to be overexpressed both in KIRC tumor tissue and in our RCC CSCs compared to normal tissue and the adherent cell lines, respectively. Furthermore, EGCG exhibits remarkable anti-tumor effects. It inhibits the proliferation and induces apoptosis of multiple myeloma cells via inactivating EZH2 [15]. Moreover, the inhibitory effect of EGCG on the EZH2 expression was confirmed experimentally, and implies a better prognosis in this study. Finally, the potential pharmacological mechanisms between ECGC and EZH2 were successfully predicted to further confirm the network between ECGC and EZH2. Combined with these factors, confirmation of the anti-tumor role of EGCG in targeting EZH2 overexpression in RCC may help us to target the CSCs and achieve a better clinical prognosis in the future.
- Which biomarkers are closely associated with the RCC cells migration and invasion?
Generally, for the migration, the densities of F-actin and integrin showed linear correlations with cellular elasticity and adhesion strength and were considered as biomarkers to quantify cell migration characteristics [16]. As for the invasion, several biomarkers can be used such as proteins in extracellular matrix, plasma proteins, membrane proteins, cytoplasm/cytosol, cytoskeleton, endoplasmic reticulum, mitochondrion and nucleus . Moreover, miR-145 is regarded as a potential biomarker of cancer migration and invasion[18] including RCC[19] . miR-101-3p serves as a tumor suppressor for RCC and inhibits its invasion and metastasis by targeting EZH2[20] .
- Authors indicated that SKRC-17 sphere and RCC-53 sphere groups were less sensitive to EGCG than the adherent SKRC-17 and RCC-53 cells. Why?
CSCs are a population of cells with self-renewal capacity and with a highly tumorigenic potential, which can adopt easily with changes in the nearby milieu. Using the conventional treatment strategies, the CSCs are more difficult to eradicate than the more differentiated cells within a tumor. The remaining CSCs are able to generate again the bulk of tumor, which results in chemotherapy resistance and causes cancer recurrence [21,22]. Based on the results of our study, sphere cell lines were less sensitive to EGCG than the adherent cells, which confirmed the resistant character of CSCs and the importance of a CSC-targeting treatment strategy in RCC.
- Authors measured the effect of EGCG on migration and invasion of SKRC[1]17, RCC-53 adherent cell lines. However, the expression of MMPs and TIMPs should be compared.
This is a good suggestion. The tissue inhibitors of metalloproteinases (TIMPs), which inhibit matrix metalloproteinases (MMPs) as well as the closely related, disintegrin-metalloproteinases (ADAMs) and ADAMs with thrombospondin motifs (ADAMTSs), which were traditionally thought to control extracellular matrix (ECM) proteolysis through direct inhibition of MMP-dependent ECM proteolysis, play central roles in morphogenesis, wound healing, tissue repair and remodelling in response to injury [23]. Based on those studies a large number of MMP inhibitors have been designed and synthesized and some were clinically tested for the treatment of patients with cancer or arthritis, but they showed little efficacy [24]. Regarding our RCC study, the assessment of changes in both the TIMPs and MMPs levels and the proteolytic function of active MMPs in the tumor stroma might be important for predicting ECM remodeling and RCC metastatic potential [25]. We will study the effect of EGCG on TIMPs and MMPs in RCC in future experiments. At present, our reported effects on migration and invasion, which are both features of aggressiveness, are a convincing first indication that EGCG treatment might effectively combat RCC aggressiveness.
References (1 – 14) are now part of the manuscript (references 61 – 74).
- Fioravanti, R.; Stazi, G.; Zwergel, C.; Valente, S.; Mai, A. Six years (2012–2018) of researches on catalytic ezh2 inhibitors: the boom of the 2‐pyridone compounds. The Chemical Record 2018, 18, 1818-1832.
- Wang, S.; Zhu, Y.; He, H.; Liu, J.; Xu, L.; Zhang, H.; Liu, H.; Liu, W.; Liu, Y.; Pan, D. Sorafenib suppresses growth and survival of hepatoma cells by accelerating degradation of enhancer of zeste homolog 2. Cancer Science 2013, 104, 750-759.
- Bao, B.; Ali, S.; Banerjee, S.; Wang, Z.; Logna, F.; Azmi, A.S.; Kong, D.; Ahmad, A.; Li, Y.; Padhye, S. Curcumin Analogue CDF Inhibits Pancreatic Tumor Growth by Switching on Suppressor microRNAs and Attenuating EZH2 ExpressionTargeting miRNA-Mediated Inactivation of EZH2 by CDF. Cancer Research 2012, 72, 335-345.
- Hua, W.-F.; Fu, Y.-S.; Liao, Y.-J.; Xia, W.-J.; Chen, Y.-C.; Zeng, Y.-X.; Kung, H.-F.; Xie, D. Curcumin induces down-regulation of EZH2 expression through the MAPK pathway in MDA-MB-435 human breast cancer cells. European Journal of Pharmacology 2010, 637, 16-21.
- Dimri, M.; Bommi, P.V.; Sahasrabuddhe, A.A.; Khandekar, J.D.; Dimri, G.P. Dietary omega-3 polyunsaturated fatty acids suppress expression of EZH2 in breast cancer cells. Carcinogenesis 2010, 31, 489-495.
- Zhang, J.-g.; Guo, J.-F.; Liu, D.-L.; Liu, Q.; Wang, J.-J. MicroRNA-101 exerts tumor-suppressive functions in non-small cell lung cancer through directly targeting enhancer of zeste homolog 2. Journal of Thoracic Oncology 2011, 6, 671-678.
- Ougolkov, A.V.; Bilim, V.N.; Billadeau, D.D. Regulation of pancreatic tumor cell proliferation and chemoresistance by the histone methyltransferase enhancer of zeste homologue 2. Clinical Cancer Research 2008, 14, 6790-6796.
- De Carvalho, D.; Binato, R.; Pereira, W.d.O.; Leroy, J.M.G.; Colassanti, M.; Proto-Siqueira, R.; Bueno-Da-Silva, A.; Zago, M.A.; Zanichelli, M.; Abdelhay, E. BCR–ABL-mediated upregulation of PRAME is responsible for knocking down TRAIL in CML patients. Oncogene 2011, 30, 223-233.
- Avan, A.; Crea, F.; Paolicchi, E.; Funel, N.; Galvani, E.; Marquez, V.E.; Honeywell, R.J.; Danesi, R.; Peters, G.J.; Giovannetti, E. Molecular Mechanisms Involved in the Synergistic Interaction of the EZH2 Inhibitor 3-Deazaneplanocin A with Gemcitabine in Pancreatic Cancer CellsDZNeP/Gemcitabine Combination in Pancreatic Cancer. Molecular Cancer Therapeutics 2012, 11, 1735-1746.
- Fiskus, W.; Rao, R.; Balusu, R.; Ganguly, S.; Tao, J.; Sotomayor, E.; Mudunuru, U.; Smith, J.E.; Hembruff, S.L.; Atadja, P. Superior Efficacy of a Combined Epigenetic Therapy against Human Mantle Cell Lymphoma CellsCombined Epigenetic Therapy against Human MCL Cells. Clinical Cancer Research 2012, 18, 6227-6238.
- Hayden, A.; Johnson, P.W.; Packham, G.; Crabb, S.J. S-adenosylhomocysteine hydrolase inhibition by 3-deazaneplanocin A analogues induces anti-cancer effects in breast cancer cell lines and synergy with both histone deacetylase and HER2 inhibition. Breast Cancer Research Treatment 2011, 127, 109-119.
- Sun, F.; Chan, E.; Wu, Z.; Yang, X.; Marquez, V.E.; Yu, Q. Combinatorial pharmacologic approaches target EZH2-mediated gene repression in breast cancer cells. Molecular Cancer Therapeutics 2009, 8, 3191-3202.
- Lv, Y.; Yuan, C.; Xiao, X.; Wang, X.; Ji, X.; Yu, H.; Wu, Z.; Zhang, J. The expression and significance of the enhancer of zeste homolog 2 in lung adenocarcinoma. Oncology Reports 2012, 28, 147-154.
- Hu, S.; Yu, L.; Li, Z.; Shen, Y.; Wang, J.; Cai, J.; Xiao, L.; Wang, Z. Overexpression of EZH2 contributes to acquired cisplatin resistance in ovarian cancer cells in vitro and in vivo. Cancer Biology Therapy 2010, 10, 788-795.
- Zhou, C.; Hui, L.; Luo, J. Epigallocatechin gallate inhibits the proliferation and induces apoptosis of multiple myeloma cells via inactivating EZH2. European Review for Medical and Pharmacological Sciences 2018, 22, 2093-2098.
- Kwon, S.; Yang, W.; Moon, D.; Kim, K.S.J.Biomarkers to quantify cell migration characteristics. Cancer Cell International 2020, 20, 1-13.
- Pouliquen, D.L.; Boissard, A.; Coqueret, O.; Guette, C.J.I.Biomarkers of tumor invasiveness in proteomics. Interrnational Journal of Oncology 2020, 57, 409-432.
- Xu, W.-X.; Liu, Z.; Deng, F.; Wang, D.-D.; Li, X.-W.; Tian, T.; Zhang, J.; Tang, J.-H. MiR-145: a potential biomarker of cancer migration and invasion. American Journal of Translational Research 2019, 11, 6739.
- Doberstein, K.; Steinmeyer, N.; Hartmetz, A.-K.; Eberhardt, W.; Mittelbronn, M.; Harter, P.N.; Juengel, E.; Blaheta, R.; Pfeilschifter, J.; Gutwein, P. MicroRNA-145 targets the metalloprotease ADAM17 and is suppressed in renal cell carcinoma patients. Neoplasia 2013, 15, 218-IN231.
- Dong, Y.; Gao, Y.; Xie, T.; Liu, H.; Zhan, X.; Xu, Y. miR-101-3p Serves as a Tumor Suppressor for Renal Cell Carcinoma and Inhibits Its Invasion and Metastasis by Targeting EZH2. BioMed Research International 2021, 2021.
- Abbaszadegan, M.R.; Bagheri, V.; Razavi, M.S.; Momtazi, A.A.; Sahebkar, A.; Gholamin, M. Isolation, identification, and characterization of cancer stem cells: A review. Journal of Cellular Physiology 2017, 232, 2008-2018.
- Chan, M.M.; Chen, R.; Fong, D. Targeting cancer stem cells with dietary phytochemical-repositioned drug combinations. Cancer Letters 2018, 433, 53-64.
- Nagase, H.; Visse, R.; Murphy, G. Structure and function of matrix metalloproteinases and TIMPs. Cardiovascular Research 2006, 69, 562-573.
- Baker, A.H.; Edwards, D.R.; Murphy, G. Metalloproteinase inhibitors: biological actions and therapeutic opportunities. Journal of Cell Science 2002, 115, 3719-3727.
- Hashmi, F.; Mollapour, M.; Bratslavsky, G.; Bourboulia, D. MMPs, tyrosine kinase signaling and extracellular matrix proteolysis in kidney cancer. In Proceedings of the Urologic Oncology: Seminars and Original Investigations, 2021; pp. 316-321.
Round 2
Reviewer 2 Report
This paper clearly revised according to the reviewer's comments.